# Gastrodin Ameliorates Cognitive Dysfunction in Vascular Dementia Rats by Suppressing Ferroptosis via the Regulation of the Nrf2/Keap1-GPx4 Signaling Pathway

**DOI:** 10.3390/molecules27196311

**Published:** 2022-09-24

**Authors:** Yue Li, Erdong Zhang, Hong Yang, Yongxin Chen, Ling Tao, Yini Xu, Tingting Chen, Xiangchun Shen

**Affiliations:** 1Guiyang Maternal and Child Health-Care Hospital, Guiyang 550002, China; 2The State Key Laboratory of Functions and Applications of Medicinal Plants, Guizhou Medical University, University Town, Guiyang 550025, China; 3The High Efficacy Application of Natural Medicinal Resources Engineering Center of Guizhou Province, the High Educational Key Laboratory of Guizhou Province for Natural Medicinal Pharmacology and Druggability, Guizhou Medical University, University Town, Guiyang 550025, China

**Keywords:** gastrodin, vascular dementia, ferroptosis, functional food, Nrf2/Keap1-GPx4

## Abstract

*Gastrodia elata* Bl. has a long edible history and is considered an important functional food raw material. Gastrodin (GAS) is one of the main functional substances in *G**. elata* BI. and can be used as a health care product for the elderly to enhance resistance and delay aging. This study investigated the ameliorative effect and mechanism of GAS on cognitive dysfunction in vascular dementia (VaD) rats, which provides a theoretical basis for development and utilization of functional food. The water maze test shows that GAS improves learning and memory impairment in VaD rats. Meanwhile; GAS significantly decreased the levels of Fe^2+^ and malondialdehyde (MDA); increased the content of glutathione (GSH); and significantly up-regulated the expression of nuclear factor erythroid 2-related factor 2 (Nrf2) and glutathione peroxidase 4 (GPx4), the key regulatory factors of ferroptosis; while it down-regulated the expression of kelch-like ECH-associated protein (Keap1) and cyclooxygenase 2 (COX2). However, GAS does not directly regulate GPx4 and COX2 to inhibit ferroptosis. Furthermore, compared with GAS alone, GAS combined with Bardoxolone (an agonist of Nrf2) did not further affect the increase in GPx4 levels and decrease in COX2 levels, nor did it further affect the regulation of GAS on the biochemical parameters of ferroptosis in HT22 hypoxia injury. These findings revealed that GAS inhibited ferroptosis in hippocampal neurons by activating the Nrf2/Keap1-GPx4 signaling pathway, suggesting its possible application as a functional food for improving vascular dementia by inhibiting ferroptosis.

## 1. Introduction

Vascular dementia (VaD) is a severe cognitive impairment syndrome caused by ischemic stroke, hemorrhagic stroke and cerebrovascular disease causing hypoperfusion in the brain region [1]. The incidence of VaD is second only to Alzheimer’s disease and is the second leading cause of dementia, which has seriously affected the normal lives of the affected population and increased the burden on families and society [2]. VaD can be prevented because preventive measures such as control of vascular risk factors, antiplatelet therapy and anticoagulation therapy have been established for pathogenic cerebrovascular diseases, but there is no symptomatic treatment for vascular dementia [3]. Therefore, understanding its basic mechanism and model is very important for the treatment of VaD.

Ferroptosis is a novel type of cell death that is distinguished from the traditional cell death pathways such as apoptosis, proptosis, necrosis and autophagy in terms of morphology, biochemistry and genetics [4]. The main features of ferroptosis are the iron accumulation and lipid peroxidation [5]. The regulation mechanism of ferroptosis involves glutathione metabolism, lipid peroxidation reactions and iron metabolism, which are closely related to the pathological process of aging, neurodegenerative diseases and ischemia reperfusion injury, and cardiovascular and cerebrovascular diseases [6,7]. It is known that ferroptosis is involved in the development of VaD, and it has gradually become a research hotspot [6]. Iron metabolism, glutathione depletion, reactive oxygen species (ROS) overproduction and marked lipid peroxidation were significantly altered in neurons administered ferroptosis inducers, confirming neuronal susceptibility to iron-dependent ferroptotic cell death [8]. Therefore, lipid peroxidation and oxidative stress are important factors in the damage of neurons in patients with VaD [9]. Recent research by Tuo Q. Z. et al. has shown that neuronal ferroptosis is closely related to ischemic stroke and cerebral hemorrhage, and ferroptosis inhibitors can significantly reduce neuron degeneration and improve neurological deficits caused by ischemic stroke and cerebral hemorrhage [10]. In addition, nuclear factor erythroid 2-related factor 2 (Nrf2) and glutathione peroxidase 4 (GPx4) are key regulators of ferroptosis. Abnormal iron metabolism and lipid peroxidation caused by the deficiency can lead to cell death, which can be seen in neurodegenerative diseases to varying degrees [11]. These studies suggest that ferroptosis mediated by NRF2-GPx4 pathway may be a crucial pathogenic factor for the occurrence and development of VaD. However, the role of ferroptosis in VaD and its specific molecular mechanism was not clarified.

*Gastrodia elata* Bl. is a common anti-aging health food for the middle-aged and the elderly, of which the main function is to relieve the tension on brain nerves and cardiovascular and cerebrovascular systems [12]. The study on the health care components and functions of *G elata* Bl. is particularly important and has received more and more attention [13]. Gastrodin (GAS) is a phenolic glycoside compound derived from the rhizome of *G elata* Bl. that can reduce ischemic brain injury by inhibiting excitotoxicity, anti-oxidation, anti-inflammatory, cell apoptosis, maintaining glial cell homeostasis, promoting neurogenesis, and protecting the integrity of the blood–brain barrier [14]. A recently published study revealed that GAS can fully exert antioxidant effects; effectively remove lipid peroxides in serum, brain and other tissues; and improve the activity of various antioxidant enzymes, thereby, exerting neuroprotective effects [15]. Our previous findings suggest that GAS can improve learning and memory disorders in VaD rats [16]. In particular, we and other researchers have found that GAS can ameliorate oxidative stress and ferroptosis through the Nrf2/HO-1 signal pathway [17]. Additionally, GAS up-regulates the level of Nrf2 and increases the expression of downstream antioxidant stress factors, such as heme oxygenase 1 (HO-1) and GPx4 [18].Our study also confirmed the effect of GAS on learning and memory impairment. Interestingly, ferrostatin-1 also significantly repaired cognitive dysfunction in VaD rats, suggesting that inhibition of ferroptosis can improve neural discrimination and cognitive decline. Therefore, GAS can be used as an additive to prevent and treat cognitive dysfunction by inhibiting neuronal ferroptosis and become the basic substance for the development of *G elata* Bl. functional food. The present study aimed to investigate whether GAS could improve neurological damage and cognitive function in VaD by reducing oxidative stress and inhibiting ferroptosis in hippocampal neurons, which provides new theoretical guidance for the optimization of clinical prevention and treatment strategies for VaD.

## 2. Results

### 2.1. GAS Significantly Alleviates Cognitive Dysfunction in Vascular Dementia rats

VaD is a type of dementia with cognitive and memory impairment as the main clinical manifestations [2]. In the present study, the Morris water maze test was used to test the learning and memory function of each group of rats, and the results showed that VaD rats established by bilateral common carotid artery occlusion (BCCAO) had learning and memory impairment. Mainly in the navigation tests, the escape latency of VaD rats was significantly longer than that of the sham group (Figure 1A). At the same time, the time passing through the target quadrant and the residence time in the target quadrant of the VaD rats were significantly shorter than the sham group in the probe trial. However, the swimming distance of VaD rats within 2 min was significantly higher than that of the sham group (Figure 1B,D,E). Compared with the rats in the VaD group, GAS treated VaD rats had shorter escape latency, longer dwell time in the original target quadrant, and significantly increased numbers for crossing the target quadrant. This indicated that GAS alleviated the cognitive impairment in VaD rats, which was consistent with the previous study in our laboratory [16]. Interestingly, a ferroptosis inhibitor (ferrostatin-1) showed the same trend as GAS to improve learning and memory impairment in VaD rats, suggesting that inhibition of ferroptosis might improve cognitive dysfunction in VaD rats.

### 2.2. GAS Up-Regulates Nrf2 Signaling Pathways and Inhibits Ferroptosis to Ameliorate Hippocampal Damage in Vascular Dementia Rats

Ferroptosis is one of the important pathogeneses of VaD, in which the levels of Fe^2+^, GSH and MDA can comprehensively reflect the level of ferroptosis [6,19]. Therefore, we confirmed the inhibitory effect of GAS on ferroptosis in VaD by evaluating the levels of Fe^2+^, GSH and MDA in hippocampal homogenates. As shown in Figure 2, the content of Fe^2+^ and MDA in the hippocampal homogenate of VaD rats was significantly increased compared with the sham group, and the content of GSH was significantly decreased, indicating abnormal iron metabolism and lipid peroxidation in the hippocampus of VaD rats (Figure 2A–C). However, GAS and ferrostatin-1 can significantly reverse the abnormal phenomenon, suggesting that GAS and ferrostatin-1 can significantly inhibit ferroptosis and improve the cognitive function of VD rat models by decreasing iron deposition, strengthening antioxidant capacity, and decreasing lipid peroxidation.

Ferroptosis is closely related to the deterioration of neurological diseases such as vascular dementia, and inhibition of ferroptosis can delay the development of vascular dementia symptoms within a certain range, thereby, reducing the impact of the disease [20]. In the present study, we aimed to further determine whether GAS could ameliorate cognitive impairment in VaD rats by inhibiting ferroptosis. The protein expression of GPx4 (a central regulator of ferroptosis) was significantly decreased and the expression of COX2 (an enzyme responsible for ferroptosis-sensitive phospholipid biosynthesis) was significantly increased in the hippocampus of VaD rats (Figure 2F,G). However, the abnormal GPx4 decrease and COX2 increase in the VaD rat hippocampus were reversed by GAS, which suggested that GAS improves hippocampal injury in VaD rats by inhibiting ferroptosis. Nrf2 transcription factor plays a protective role in degenerative diseases by inducing multiple cellular defenses, regulating the expression of detoxification enzymes [11]. Nrf2 and GPx4 are key regulators of ferroptosis, and down-regulation of Nrf2 and GPx4 can lead to imbalance in intracellular oxidation and antioxidant systems, resulting in intracellular ROS increase and cell death [21]. Conversely, activation of Nrf2 promotes iron storage, reduces cellular iron uptake and limits the production of reactive oxygen species [22]. To determine whether GAS inhibits ferroptosis in VaD rat hippocampus by regulating Nrf2 signaling pathway, we detected the protein expression of Nrf2 and Keap1 by Western blot assay. Our experimental study found that the protein expression of Nrf2 in the hippocampus of VaD rats treated with GAS or ferrostatin-1 was higher than that of the VaD model group, and the protein expression of Keap1 was lower than that of the VaD model group (Figure 2D,E). These results indicate that GAS may inhibit ferroptosis by regulating Nrf2 signaling, thereby, improving learning and memory dysfunction in VaD rats.

### 2.3. GAS Inhibits Hypoxia-Induced Ferroptosis in HT22 Cells 

GPx4 is a key regulatory factor of glutathione dependence in the ferroptosis regulation pathway, it can protect cells from accumulation of lipid peroxide by removing free radicals and participating in the hydrolysis of lipid peroxide [23]. Glutathione deletion or inhibition of GPx4 expression can promote ferroptosis [24]. First, as shown in Appendix A, When the HT22 cells were hypoxic for 36 h, the survival rate was significantly reduced, and GAS had a significant improvement effect on the hypoxia-induced decrease in the survival rate of HT22 cells. Furthermore, when HT22 cells were cultured in a hypoxic incubator with 5% O_2_ for 36 h to establish a VaD cell model, we determined changes in the levels of ferroptosis-related markers and observed significant up-regulation of COX2 protein levels, as well as down-regulation of GPx4 (Figure 3A). In addition, the accumulation of ROS was increased, the synthesis of GSH was blocked, and the intracellular Fe^2+^ content was also significantly increased (Figure 3C–E), suggesting that hypoxia induces ferroptosis in HT22 cells. However, the above phenomenon can be reversed after treatment of cells with GAS, as shown in Figure 3. These results suggest that co-treatment with GAS reversed hypoxia-induced changes in ferroptosis-related markers in HT22 cells.

### 2.4. GAS Inhibits Hypoxia-Induced Ferroptosis in HT22 Cells and Promotes GPx4 Signaling Pathway

GPx4 is a key antioxidant enzyme that negatively regulates neuronal ferroptosis by reducing intracellular lipid peroxidation, while GXP4 ablation induces ferroptosis and triggers neurodegeneration in forebrain neurons [23]. Herein, as shown in Figure 4, the effect of GAS on ferroptosis was evaluated in combination with ferroptosis inducers (erastin). In HT22 cells treated with erastin alone, the protein expression of GPx4 and the activity of GSH decreased, the protein expression of COX2 and the level of ROS increased (Figure 4A). Moreover, hypoxia-induced ablation of GPx4 and GSH and accumulation of COX2, Fe^2+^ and ROS further increased after erastin treatment (Figure 4C–E). Pre-incubation with GAS attenuated hypoxia-induced ablation of GPx4 and GSH and accumulation of COX2, ROS and Fe^2+^, suggesting that GPx4 and COX2 are involved in hypoxia-induced ferroptosis in HT22 cells, while GAS inhibits HT22 cell ferroptosis by regulating GPx4 and COX2. These GAS effects were further confirmed by examining the impact of ferrostatin-1, a ferroptosis inhibitor, on hypoxia-induced GPx4 ablation and COX2 accumulation. As expected, ferrostatin-1 alone attenuated hypoxia-induced COX2 increase and GPx4 ablation, with further attenuation observed when GAS was combined with ferrostatin-1 (Figure 4B).

### 2.5. GAS Promotes Nrf2 Nuclear Transfer to Alleviate Hypoxic Injury in HT22 Cells

Transcriptional activation of Nrf2 is associated with anti-ferroptosis [11]. In the ferroptosis pathway, most of the cascaded or interacting enzymes and proteins, such as GPx4, HO-1, etc., are transcribed by the antioxidant response element Nrf2 adjust [21]. Inactivation, inhibition and knockdown of the Nrf2 gene all enhance cellular ferroptosis [25]; while activation of the Nrf2 signaling pathway to reduce ferroptosis can improve neurodegenerative diseases [11]. In the present study, we aimed to investigate the effect of GAS on Nrf2 signaling upon hypoxic injury in HT22 cells. As shown in Figure 5, the protein expression of Nrf2 was significantly reduced when HT22 cells were injured by hypoxia, suggesting that its nuclear transfer was reduced, and its antioxidant defense effect was weakened (Figure 5A). Similarly, immunohistochemical experiments confirmed this result (Figure 5B). Nrf2 is ubiquitinated after binding to its negative regulator kelch-like ECH-associated protein 1 (Keap1) in the cytoplasm [26]. In this experiment, we also found that the expression of Keap1 was significantly up-regulated by HT22 during hypoxic injury. In contrast, GAS can promote the nuclear translocation of Nrf2, reduce the protein expression of Keap1, and up-regulate the expression level of antioxidant response genes, thereby, increasing the ability of cells to resist oxidative stress.

### 2.6. GAS Inhibits Hypoxia-Induced Ferroptosis in HT22 Cells via Nrf2-GPx4 Signaling Pathway

Nrf2 and GPx4 are key regulators of ferroptosis, and their down-regulation can cause an imbalance of intracellular oxidative and antioxidant systems, resulting in increased intracellular ROS, which in turn leads to cell death [11]. Accordingly, we further explored the mechanism through which GAS attenuates hypoxia-induced ferroptosis by promoting the Nrf2 signaling pathway in HT22 cells. As shown in Figure 6A, the inhibitor of Nrf2 (ML385) significantly reduced the protein expression of GPx4 and increased the protein expression of COX2, suggesting that inhibition of Nrf2 activity would cause ferroptosis in HT22 cells. However, under hypoxia, GAS could reverse the ML385-induced decrease in GPx4 protein expression and increase in COX2 expression. The above results suggest that GAS may inhibit hypoxia-induced ferroptosis by promoting the transcriptional activity of Nrf2. To further confirm the role of Nrf2 in GAS-mediated inhibition of ferroptosis, an agonist of Nrf2 (Bardoxolone) alone was administered to HT22 cells under hypoxic conditions for 36 h. The results showed that GPx4 protein expression was significantly up-regulated, GSH level was significantly increased, COX2 protein expression was significantly down-regulated (Figure 6B,D), the intracellular Fe^2+^ content was significantly decreased, and ROS accumulation was also decreased (Figure 6C,E). However, when GAS and bardoxolone was combined on HT22 cells, there was no further change in GPx4 and COX2 and no further changes in the biochemical indicators of ferroptosis (Figure 6B). Collectively, these findings suggest that GAS may inhibit hypoxia-induced ferroptosis in HT22 cells by activating the Nrf2/Keap1-GPx4 signaling pathway, as shown in Figure 7.

## 3. Discussion

VaD is an age-related neurodegenerative disease characterized by cognitive impairment caused by chronic cerebral blood flow, and it is the second most common neurodegenerative disease after Alzheimer’s disease (AD) [1]. Our previous findings suggest that GAS can improve learning and memory disorders in VaD rats [16]. Our study also confirmed the effect of GAS on learning and memory impairment. Interestingly, ferrostatin-1 also significantly repaired cognitive dysfunction in VaD rats, suggesting that inhibition of ferroptosis can improve neural discrimination and cognitive decline.

Ferroptosis is a form of iron-dependent regulation of cell death, which is mainly manifested by strong oxidative stress, membrane lipid peroxidation and obvious cell morphological changes [8]. Iron and lipid peroxides are major players in the ferroptosis process [27]. In addition, the brain is an organ in which iron tends to accumulate gradually with age, and iron accumulation in the brain can easily lead to the production of a large number of oxidative free radicals, peroxides, etc., thereby, causing lipid peroxidation and protein function disorders and eventually leading to neuronal degeneration and death [28]. Other studies have shown that many small-molecule ferroptosis inhibitors can improve neurodegeneration and cognitive decline and have potential neuroprotective abilities [29], suggesting that ferroptosis may be a potential neurodegenerative mechanism and has certain effects on neurons involved in learning and memory. Our experimental study discovered a promotion of Fe^2+^ levels, a decrease in GSH synthesis and an accumulation of ROS and peroxidation product MDA. The above results suggest that iron accumulation, decreased antioxidant capacity and lipid peroxidation in VaD model rats induce ferroptosis in hippocampal neurons, which are involved in the occurrence and development of cognitive dysfunction in VaD. Under stress conditions, Fe^3+^ is reduced to Fe^2+^ intracellularly. The excessive accumulation of Fe^2+^ in cells will promote the generation of oxygen free radicals and lipid peroxidation product MDA through the Fenton reaction, which destroys macromolecules such as proteins and nucleic acids and leads to ferroptosis in nerve cells, which causes cognitive decline [30]. In this study, it was found that GAS can decrease the accumulation of Fe^2+^ and ROS, reduce the production of MDA and increase the content of GSH, suggesting that GAS down-regulates the transport capacity of intracellular iron ions, thereby, inhibiting the excessive accumulation of intracellular iron ions and the generation of lipid peroxides during ferroptosis in VaD rats.

GPx4 is an antioxidant similar to other selenium-containing GPx enzymes. Four isoforms of GPx4 were found, namely, cGPx4, mGPx4, snGPx4 and GPx4-I. Among them, cGPx4 is ubiquitous in cells and is the main subtype of neurons, which can be observed in both cell membrane and cytoplasm [24]. GPx4 converts reduced glutathione to oxidized glutathione (GSSG) and reduces cytotoxic lipid peroxide (L-OOH) to the corresponding alcohol (L-OH) or free hydrogen peroxide converted into water. Inhibition of GPx4 activity results in the inability of sensitive cells to clear accumulated lipid peroxides, resulting in ferroptosis [31]. In mice, GPx4 knockdown causes age-related neurodegenerative changes and neuronal loss [32].In addition, promoting the expression of the lipid antioxidant GPx4 prevents neuronal damage in a hemorrhagic stroke model [33]. Our results showed that GPx4 was significantly inhibited in the hippocampus of VaD rats, and the same results were observed in hypoxia-induced HT22 cell injury. However, GAS can effectively increase the level of GPx4 in the hippocampus of VaD rats, inhibit the degree of ferroptosis in the hippocampus and reduce the damage of hippocampal neurons. It, therefore, appears that increasing the activity of GPx4, promoting the antioxidant capacity of cells, reducing the accumulation of lipid peroxides and, thus, inhibiting ferroptosis becomes an effective way to improve VaD. Furthermore, GAS and GAS combined with ferrostatin-1, a ferroptosis inhibitor, increased the levels of GPx4 in HT22 cells. Meanwhile, we observed that erastin (a ferroptosis inducer) alone decreased the expression of GPx4 and promoted the expression of COX2. When erastin acted on HT22 cells under hypoxia, we observed further down-regulation of GPx4 expression and up-regulation of COX2 expression; while GAS promotes the expression of GPx4 and reduces the expression of COX2. These results suggest that GAS may regulate the activity changes of GPx4 and COX2 through other upstream signals, thereby inhibiting the occurrence of ferroptosis in VaD hippocampal neurons. It was proposed that Nrf2 can regulate ferroptosis to play a neuroprotective role by regulating the transcription of its downstream genes [34]. Therefore, further investigation is needed to explore the detailed mechanism of the action of GAS to inhibit ferroptosis in VaD hippocampal neurons.

Nrf2 is the key regulatory factor required for cells to maintain an oxidative steady state and is activated under conditions of high oxidative stress [11]. When cells are subjected to a large amount of stress, the cysteine residues on the Keap1 structure bind to intracellular electrophiles, which leads to the conformational change of Keap1, which leads to the dissociation of Nrf2 and Keap1 and the transfer of Nrf2 into the nucleus [26]. Previous observations have shown that lack of Nrf2 significantly exacerbates cognitive deficits in an APP/PS1 mouse model of AD; however, activation of Nrf2 to increase antioxidant protein levels is an effective method of neuroprotection [35]. In agreement with these observations, our data revealed that the accumulation of Keap1 and loss of Nrf2 in the hippocampus of VaD rats were reversed by GAS, which suggests that the protective effect of GAS on learning and memory in VaD rats is related to the activation of the Nrf2 pathway to inhibit oxidative stress. The activation of Nrf2 promotes the storage of iron, reduces the absorption of iron and limits the production of reactive oxygen species [11]. In addition, Nrf2 and GPx4 are key regulators of ferroptosis, and down-regulation of Nrf2 and GPx4 can lead to an imbalance in intracellular oxidation and antioxidant systems, resulting in intracellular ROS increase and cell death [25]. A recent study reports that targeting the Nrf2/GPx4 axis effectively modulates ferroptosis [36]. Jiang T et al. found that GAS protects HT22 cells from glutamate-induced ferroptosis via the Nrf2/HO-1 signaling pathway [17]. Therefore, further studies are needed to explore the detailed mechanism of the action of GAS to inhibit ferroptosis by activating the Nrf2 signaling pathway. Herein, we observed that ML385, a Nrf2 inhibitor, alone decreased the expression of GPx4 and promoted the expression of COX2, whereas treatment with GAS in combination with ML385 did reverse this change in HT22 cells. To further confirm the role of Nrf2 in GAS-mediated inhibition of ferroptosis, an agonist of Nrf2 (Bardoxolone) alone was administered to HT22 cells under hypoxic conditions for 36 h and the results showed that the protein expression of GPx4 was significantly up-regulated, and the protein expression of COX2 was significantly down-regulated. However, treatment with GAS in combination with bardoxolone did not reverse this change in HT22 cells. Our findings confirm that GAS significantly reduced the level of ferroptosis through activating the Nrf2-Keap1-GPx4 pathway in vivo and in vitro.

Collectively, our data suggest that ferroptosis is involved in the occurrence and development of VaD. GAS inhibits ferroptosis in hippocampal neurons by activating the Nrf2/Keap1-GPx4 signaling pathway to decrease iron deposition, strengthen antioxidant capacity and decrease lipid peroxidation. Our research suggests that the development of functional food for *G. elata* BI. is based on GAS, and thus, dietary intake of GAS can be used to prevent and treat behaviors such as memory loss and learning deficits associated with neurodegenerative diseases. Therefore, this study provides a scientific and technological basis for the development and utilization of GAS functional food.

## 4. Materials and Methods

### 4.1. Reagent and Antibodies

GAS (SMB00313) was purchased from Sigma-Aldrich (St Louis, MO, USA). Ferroptosis inhibitor (Ferrostatin-1, HY-100579), ferroptosis inducers (Erastin, HY-15763), ML385 (HY-100523), Bardoxolone (HY-14909) were purchased from MedChemExpress. Primary antibodies against β-actin (66009-1-Ig, 1:10,000), GPx4 (67,763-1-Ig, 1:1000), COX2 (66,351-1-Ig, 1:1000), Nrf2 (80593-1-RR, 1:1000), Keap1 (60027-1-Ig, 1:1000), were purchased from Proteintech (Wuhan, China). HRP-conjugated anti-rabbit (BS13278, 1:10,000) and anti-mouse (BS12478, 1:10,000) secondary antibodies were obtained from Bioworld. Goat PAb to Rb IgG Alexa Fluor^®^488 (Ab150077, 1:200) was purchased from Abcam.

### 4.2. Animal Groups and Experimental Design

Male Sprague–Dawley rats (weight 260 ± 20 g; Guizhou Medical University Experimental Animal Center; Certificate No. SCXK2018-0001; Grant No. 2200483) were reared in a specific pathogen-free environment with 12 h light/dark cycle and 55% ± 10% humidity at a temperature of 20~25 °C, were provided with sufficient feed and sterile drinking water and fasted for 6 h before and after surgery. All animal experiments were performed in accordance with the Declaration of Helsinki and the Guide for the Care and Use of Laboratory Animals.

Except for the sham-operated group (only threading without ligation), the other groups were treated with BCCAO to establish vascular dementia models [37]. The Morris water maze test, one of the most commonly used spatial learning and memory tests in behavioral neuroscience [38], was assessed on the third day after modeling. Taking the escape latency of rats in the sham-operated group as the reference value, the ratio of the escape latency of the other groups to the reference value was calculated. If the ratio was greater than 20%, it was rated as a rat with cognitive dysfunction, that is, the model was successfully established. The model rats were randomly divided into model group, GAS low-dose group (25 mg/kg), GAS high-dose group (50 mg/kg), Ferrostatin-1 positive control group (2 mg/kg). In the sham group, the common carotid artery was isolated, and the neck incision was sutured without ligation. The rats in each group were sacrificed after 8 weeks of administration, and the hippocampus was collected.

### 4.3. Water Maze Swimming Test

After 8 weeks of administration, the rats in each group were subjected to a water maze test. The MWM-100 Morris water maze video analysis system (Chengdu Taimeng Technology, Co., Ltd, Chengdu, China) is mainly used to test the spatial learning ability and spatial memory ability of experimental animals [39]. A transparent plastic plinth was set about 1.5 cm underwater and placed in the first quadrant as a concealed platform. The escape latency was set as 120 s. If the rat searched for the platform within 120 s and stayed for 10 s, the escape latency was generated; if the rat did not search for the platform within 120 s, the escape latency was 120 s. For the first 5 days, the rats entered the water from the sides of the tanks in the two diagonal quadrants of the water maze and performed the positioning navigation test twice a day. On the 6th day, the concealed platform was removed, and water was entered from the side of the tank in the third quadrant of the water maze to conduct a space exploration test.

### 4.4. Cell Culture and Treatment

HT22 cells, a mouse hippocampal neuronal cell line, (Shanghai Zhongqiaoxinzhou Biotech; Shanghai, China; passage number: ZQ0476), were cultured in Dulbecco’s modified Eagle medium (Gibco, Thermo Fisher Waltham, CA, USA; Cat#C11995500BT) supplemented with 10% fetal bovine serum (Zhe Jiang Tianhang Biotechnology Co., LTD, Zhejiang, China; Cat#110011-8611). The cells were cultured in an incubator containing 5% O_2_ to establish a cell model of hypoxia injury. Then, the cells were treated with various concentrations of GAS (25, 50 and 100 μmol/L) or GAS (100 μmol/L) combined with ferroptosis agonist (Erastin, 20 μmol/L), ferroptosis inhibitor (Ferrostatin-1, 20 μmol/L) or combined Nrf2 inhibitor (ML385, 20 μmol/L) and Nrf2 agonist (Bardoxolone, 0.2 μmol/L). These groups were placed in a hypoxia incubator for 36 h for subsequent studies.

### 4.5. Western Blotting

The HT22 cells or the tissue from the rat hippocampus were lysed in lysis buffer (R0010, Solarbio, Beijing, China), and the lysates were centrifuged at 12,000× *g*, 4 °C for 15 min. Total proteins (25~50 μg) were separated using 12% sodium dodecyl and transferred to a polyvinylidene fluoride membrane (ISEQ00010, Merk Millipore, Tullagreen, Carrigtwohill, Cork). The membrane was incubated with corresponding antibodies, such as Nrf2, Keap1, GPX4, etc., and the results were obtained with the ChemiDoc XRS+ system and analyzed with Image Lab™ software version 5.2 (Bio-Rad, Hercules, CA, USA).

### 4.6. Immunofluorescence

Cells were incubated on six-well plates containing coverslips. After treatment, the cells were fixed with paraformaldehyde (BL539A, Biosharp, Guangzhou, China) for 10 min, permeabilized with 0.1% Triton-X100 for 10 min, and blocked with goat serum for 40 min. The coverslips were placed in a wet box for primary antibody incubation at 4 °C overnight. After washing the coverslips with PBS, fluorescent secondary antibodies were added and incubated for 1 h. The coverslips were washed with PBS buffer, DAPI (BD5010, Bioworld) was added dropwise, and the results were observed on a DMi8 fluorescence microscope.

### 4.7. GSH and MDA Detection 

MDA and GSH were measured using an MDA (Cat#BC0025, Solarbio, Beijing, China) or GSH (A006-2-1, Nanjing Jiancheng Bioengineering Institute, Nanjing, China) assay kit following the standard protocol. Briefly, the double-antibody sandwich method was used to measure the optical density of each well at a wavelength, and the contents of MDA and GSH were calculated.

### 4.8. Detection of Cellular Fe^2+^ Ions Generation 

The content of Fe^2+^ in hippocampus tissues was determined using the Fe^2+^ (A039-2-1, Nanjing Jicheng Bioengineering Institute, Nanjing, China) detection kit according to the standard manufacturer’s procedures. HT22 cells were incubated in six-well plates. After treatment, FerroOrange (F374, dojingo, Japan) with a concentration of 1 μM was prepared in serum-free medium, and then 1 mL FerroOrange was added to each well and incubated in a normal incubator (37 ℃, 21% O_2_ and 5% CO_2_) for 30 min. The results were observed on a DMi8 fluorescence microscope. The Image J software calculated relative fluorescence intensities.

### 4.9. Determination of ROS

ROS were detected using an ROS (S0033M, Beyotime, Jiangsu, China) assay kit. Detection was performed using a NovoCyte flowcytometer with NovoExpress analysis software (ACEA Biosciences, San Diego, CA, USA).

### 4.10. Statistical Analyses

All data were statistically processed and represented graphically using Graphpad Prism 8.0 software. *p* < 0.05 was indicated statistically significant. Statistical variables are expressed as the mean ± standard deviation (SD) for at least three independent experiments. A two-tailed Student’s test one-way test was used to compare the differences between the two groups, and the one-way ANOVA was used for comparison among multiple groups.

## Figures and Tables

**Figure 1 molecules-27-06311-f001:**
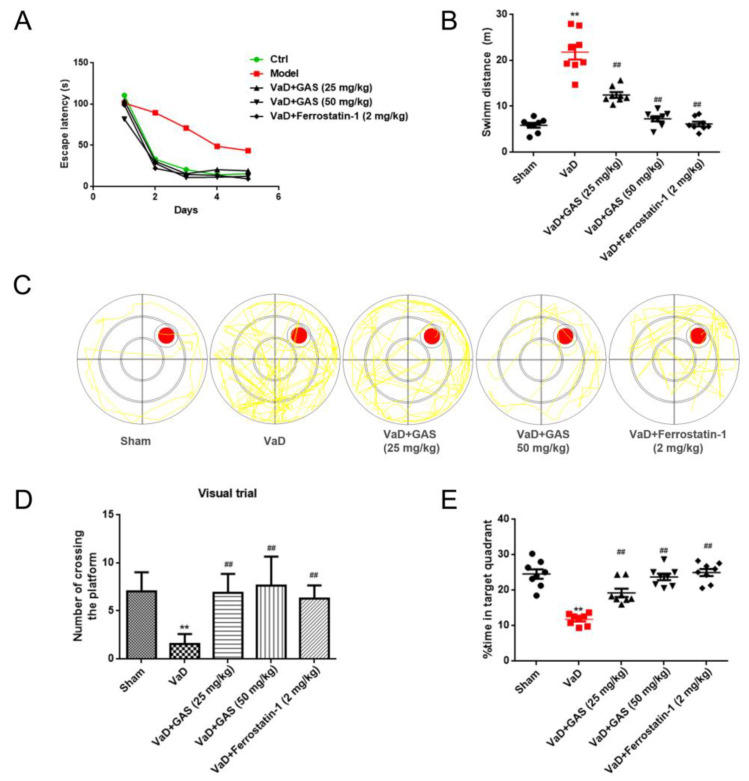
Neuroprotective effect of GAS on learning and memory impairment in VaD rats. (**A**), Escape latency (**B**), swimming distance in less than 2 min (**C**), swimming track diagram, red circle: the location of the platform; yellow line: the track of the first landing on the platform. (**D**), The number of times crossing the platform (**E**), Percentage of time spent on target project. Results are shown as mean ± SD (*n* = 6). ** *p* < 0.01 specifies the differences between sham rat and VaD rat. ^##^
*p* < 0.01 compares between VaD rat and GAS-treated rat or ferrostatin1-treated rat.

**Figure 2 molecules-27-06311-f002:**
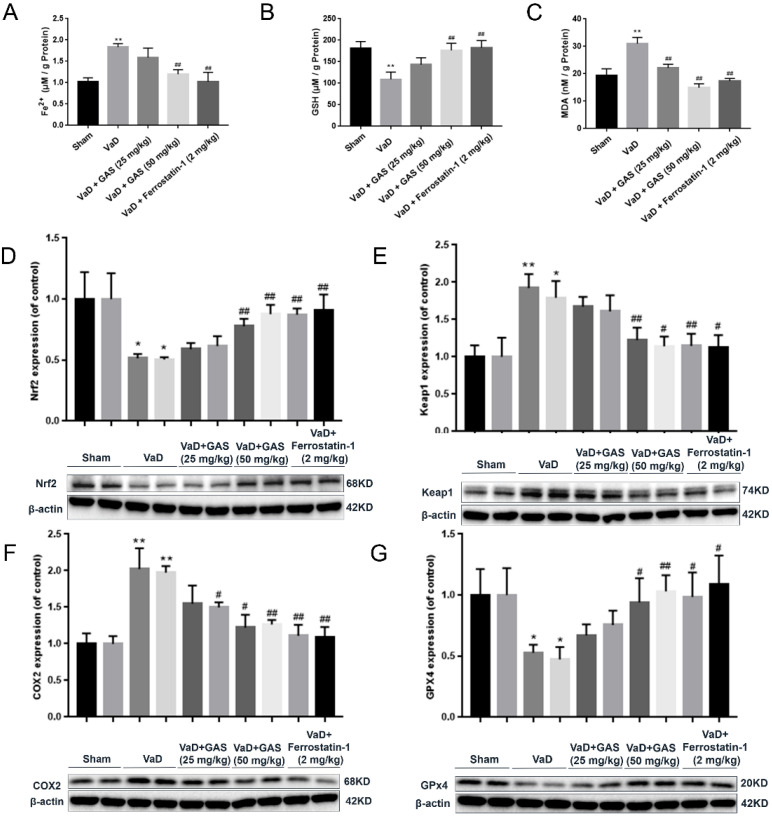
Effects of GAS on ferroptosis indexes in the hippocampus of VD rats. (**A**–**C**), detection of biochemical indicators of ferroptosis, (**D**–**G**), Western blot analyses of ferroptosis-related protein expression. Results are shown as mean ± SD (*n* = 6).* *p* < 0.05, ** *p* < 0.01 specifies the differences between sham rat and VaD rat. ^#^
*p* < 0.05, ^##^
*p* < 0.01 compares between VaD rat and GAS-treated rat or ferrostatin1-treated rat.

**Figure 3 molecules-27-06311-f003:**
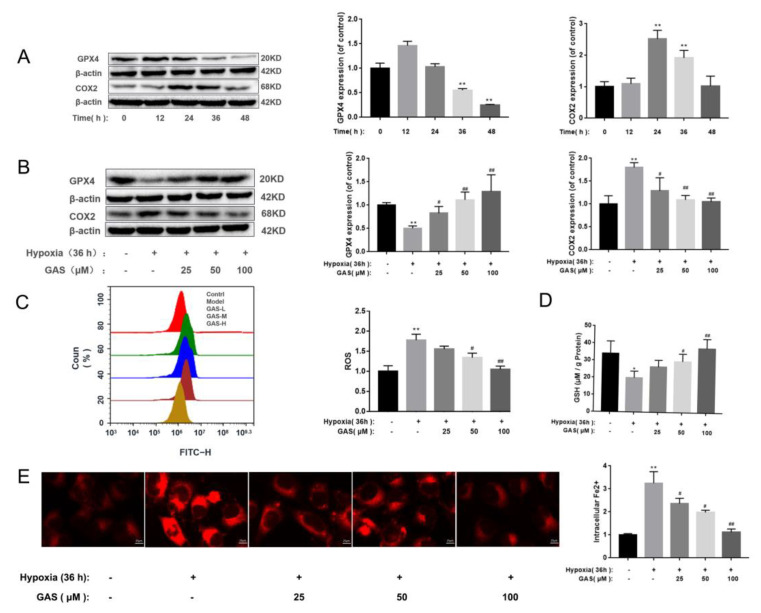
GAS inhibits hypoxia-induced ferroptosis in HT22 Cells. (**A**), Western blot showing the levels of GPx4 and COX2 at different time points of hypoxia in HT22 cells (**B**), HT22 cells were pretreated with GAS for 1 h and cultured in hypoxia for 36 h, and then, the expressions of GPx4 and COX2 were detected by Western blotting (**C**), ROS levels were measured by flow cytometry (**D**), detection of GSH content by ELISA (**E**), When HT22 cells were cultured under hypoxia for 36 h in the presence or absence of GAS, the intracellular Fe^2+^ levels were measured by immunofluorescence microscopy using FerroOrange. * *p* < 0.05, ** *p* < 0.01 versus control, *^#^ p* < 0.05, *^##^ p* < 0.01 versus hypoxia.

**Figure 4 molecules-27-06311-f004:**
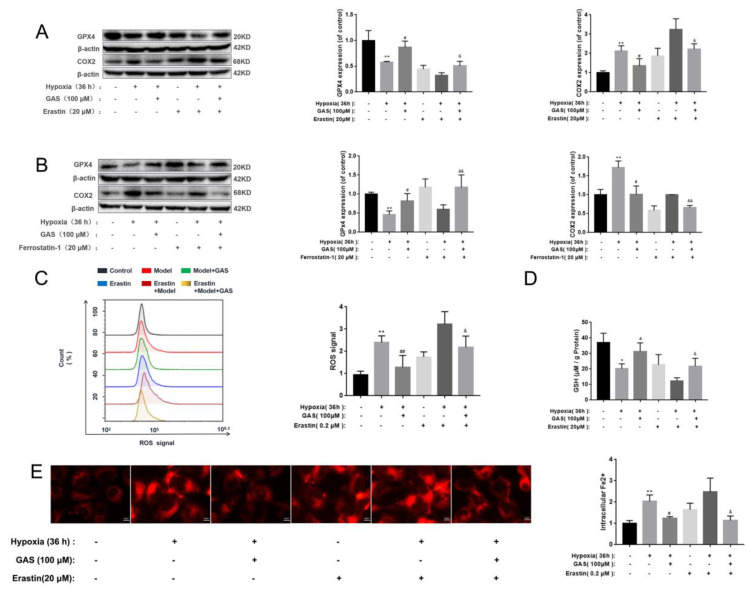
Experimental study on the improvement of hypoxic injury of HT22 cells by GAS based on the regulation of ferroptosis by the GPx4 pathway. (**A**,**B**), GPx4 and COX2 were analyzed using Western blot. Pharmacological inhibitors; erastin (ferroptosis inducers), ferrostatin-1 (ferroptosis inhibitor) (**C**), intracellular ROS levels were labeled with 2,7-Dichlorodihydrofluorescein diacetate (DCFH-DA, 10 μM) and detected by flow cytometry (*n* = 3) (**D**), levels of GSH in HT22 cells treated with ferroptosis inducers (erastin) were detected with or without GAS treatment (100 μM) for 24 h (*n* = 3) (**E**), detection of intracellular Fe^2+^ changes using FerroOrange. * *p* < 0.05, ** *p* < 0.01 compared to the control group; ^#^
*p* < 0.05, ^##^
*p* < 0.01 compared to the hypoxia group; ^&^
*p* < 0.05, ^&^^&^
*p* < 0.01, compared to the hypoxia + ferrostatin-1 or hypoxia + erastin group.

**Figure 5 molecules-27-06311-f005:**
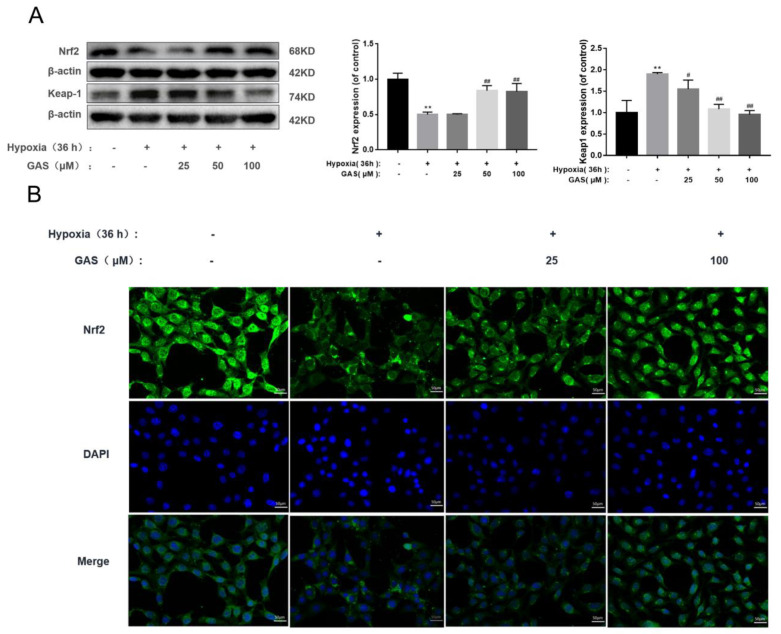
GAS activates the Nrf2 signaling pathway to ameliorate hypoxic injury in HT22 cells (**A**), HT22 cells were incubated in a normal incubator (5% CO_2_, 37 °C) with GAS for 1 h and then placed in a hypoxia incubator (5% O_2_, 5% CO_2_, 90% N_2_, 37 °C) for 36 h, and then the protein levels of GPx4 and COX2 were detected by Western blot (**B**), Immunofluorescence microscopy images of Nrf2 in HT22 after hypoxia treatment in the presence, or absence, of GAS for 36 h, and cell nuclei were stained with 4,6-diamino-2-phenyl indole (DAPI, blue fluorescence). Mean fluorescence intensity values were calculated by Image J software (*n* = 3). Magnification, ×400; scale bars, 25 µm. Results are presented as mean ± SD. ** *p* < 0.01, control versus hypoxia; ^#^
*p* < 0.05, *^##^ p* < 0.01, hypoxia versus hypoxia plus GAS.

**Figure 6 molecules-27-06311-f006:**
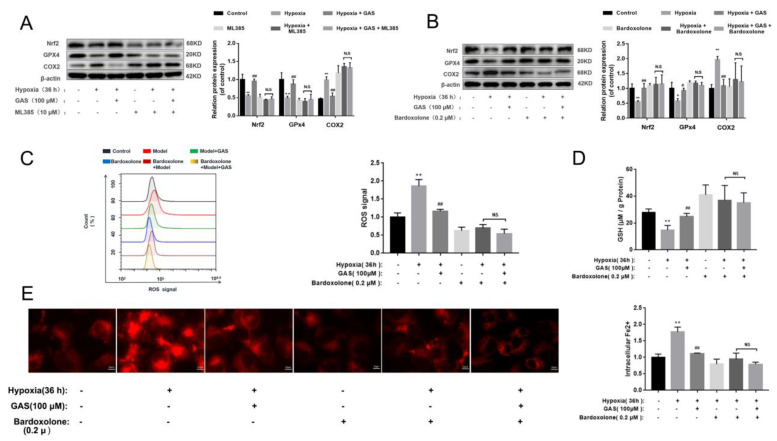
GAS inhibits ferroptosis via the Nrf2/GPx4 Pathway to improve hypoxic injury in HT22 cells. (**A**,**B**), levels of GPx4 and COX2 in HT22 cells treated with Nrf2 inhibitors (ML385) or Nrf2 agonists (Bardoxolone) were assessed with or without GAS treatment (100 μM) for 36 h (*n* = 3) (**C**), intracellular ROS levels were labeled with DCFH-DA (10 μM) and detected by flow cytometry (n = 3) (**D**), levels of GSH in HT22 cells treated with an agonist of Nrf2 (Bardoxolone) were detected with or without GAS treatment (100 μM) for 36 h (*n* = 3) (**E**), detection of intracellular Fe^2+^ changes using FerroOrange. * *p* < 0.05, ** *p* < 0.01, compared to the control group; ^#^
*p* < 0.05, *^##^ p* < 0.01 compared to the hypoxia group; NS indicates no significance.

**Figure 7 molecules-27-06311-f007:**
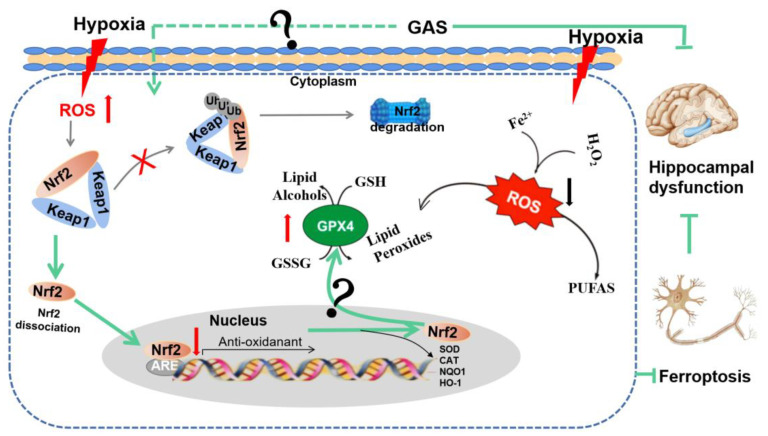
GAS ameliorates cognitive dysfunction in VaD rats by suppressing ferroptosis via the regulation of the Nrf2/Keap1-GPx4 signaling pathway. The illustration represents how the GAS up-regulates the Nrf2 gene, promotes Nrf2 release from the complex of Keap1 and Nrf2. Then, Nrf2 enters the nucleus and binds with antioxidant response elements (ARE), thereby. increasing the expression of GPx4 to decrease iron deposition, strengthen antioxidant capacity and decrease lipid peroxidation.

## Data Availability

Data analyzed or generated during this study are included in this manuscript.

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
