# Peer review of "Gastrodin Ameliorates Cognitive Dysfunction in Vascular Dementia Rats by Suppressing Ferroptosis via the Regulation of the Nrf2/Keap1-GPx4 Signaling Pathway"

_molecules, 2022, doi:10.3390/molecules27196311_

Round 1

Reviewer 1 Report

In the manuscript titled "Effect and Mechanism of Gastrodin from the edible Plant Gastrodia elata Bl on Cognitive Dysfunction in Rats", the authors evaluate the molecular mechanism of pathogenesis of Vascular dementia and show that gastrodin (GAS) ameliorates the symptoms of the disease by blocking ferroptosis in hippocampal neurons. The study is well formulated and the results support the conclusions. Below is a list of concerns that require revision:

1) Please state the full form of an acronym when it is first introduced (for example MDA)

2) Please explain the origin of HT22 cells (organism, cell type, etc) and why these cells were chosen for this study.

3) All figure legends require more details.

- Fig 1: VD and VaD are used interchangeably in the X-axis labels. Please use the same nomenclature for consistency. (B) Y axis units missing. Possible typographic error in the Y axis label. (C) Please state clearly what the red circle signifies. 

For all Western Blot and immunostaining quantifications throughout the manuscript, please state clearly how the data was normalized ( beta-actin control, initial time point, negative control condition, etc.)

- Fig 5: (B) Possible mis-annotation in the second column for GAS (Should be '-' or '0' instead of '+'.

- Fig 6: (A-B) Barplots are missing the legends to indicate what condiitons the different shades of grey correspond to.

4) Please include the name (Manufacturer and catalog number) and/or full composition of the HT22 culture medium.

5) The title of the manuscript can be improved to more accurately reflect the findings of this study.

Author Response

Point 1: Please state the full form of an acronym when it is first introduced (for example MDA).

Response 1: We apologize for our mistake. We have explained the full form of the acronym when it was first introduced and have made careful modifications on the original manuscript . All changes made to the text are in red color.

Point 2: Please explain the origin of HT22 cells (organism, cell type, etc) and why these cells were chosen for this study.

Response 2: Chronic cerebral ischemia is a progressive neurodegenerative process caused by long-term insufficient cerebral blood perfusion. It is one of the main risk factors of some neurodegenerative diseases, such as vascular dementia (VaD) and Alzheimer’s (AD). Hippocampus is one of the most important nerve centers for learning and memory and is closely related to advanced functions, such as learning, memory, and cognition. Neurons in the hippocampal CA1 area are extremely vulnerable to ischaemia or hypoxia. As a result, studying the mechanism of hippocampal neuron damage caused by VaD has important theoretical significance and application value. However, the molecular mechanism of nerve injury is not fully understood in VaD diseases. Herein, the hypoxia-induced nerve damage model was used to explore the molecular mechanisms affecting nerve damage. To investigate the positive effects of hypoxic injury on oxidative stress and ferroptosis of neurons, we simulated the ischemia process in vitro using an immortalized hippocampal neuron cell line (HT22) and cultured in an incubator containing 5% O2 to establish a hypoxic injury cell model. HT22 cells, a mouse hippocampal neuronal cell line, were purchased from Shanghai Zhongqiaoxinzhou Biotech (Shanghai, China; passage number: ZQ0476).

Point 3: All figure legends require more details

Fig 1: VD and VaD are used interchangeably in the X-axis labels. Please use the same nomenclature for consistency. (B) Y axis units missing. Possible typographic error in the Y axis label. (C) Please state clearly what the red circle signifies.

Fig 5: (B) Possible mis-annotation in the second column for GAS (Should be '-' or '0' instead of '+'.

Fig 6: (A-B) Barplots are missing the legends to indicate what condiitons the different shades of grey correspond to.

Response 3: Thank you very much. It was our mistake. We have corrected the error in figures.

Point 4: For all Western Blot and immunostaining quantifications throughout the manuscript, please state clearly how the data was normalized (beta-actin control, initial time point, negative control condition, etc.)

Response 4: Thank you for your comments. First, the gray value of each group of target proteins was divided by the gray value of the corresponding internal reference (β-actin) to correct errors. Then, the ratios of the other groups were quantified by dividing the ratio at the first time point or the ratio of the negative control. At the same time, the average value of the ratio at the first time point or the negative control ratio was taken, and then the ratio at the first time point or the negative control ratio was divided by the average value for standardization.

Point 5: Please include the name (Manufacturer and catalog number) and/or full composition of the HT22 culture medium.

Response 5: Thank you for reminding us. HT22 cells were maintained in Dulbecco’s modified Eagle medium (Gibco, Thermo Fisher Waltham, CA, USA; Cat#C11995500BT) supplemented with 10% fetal bovine serum (Zhejiang Tianhang Biotechnology Co., LTD, Zhejiang, China; Cat#110011-8611). The above content has been carried out in the revised manuscript.

Point 6: The title of the manuscript can be improved to more accurately reflect the findings of this study.

Response 6: Thank you for your suggestion.The title of the manuscript was changed to “Gastrodin Ameliorates Cognitive Dysfunction in Vascular Dementia Rats by Suppressing Ferroptosis via the Regulation of the Nrf2/Keap1-GPx4 Signaling Pathway”.

Reviewer 2 Report

The major problem of the present manuscript is its limited novelty because the main observations on the cognitive dysfunction properties of Gastrodin from the edible Plant Gastrodia elata Bl have been already reported by the same authors in previous and recent publications (see refs 16-18). However, this work reports some interesting data that may deserve publication.  

Main criticisms:

1) Lines 75-77 of the introduction: the letter is enlarged.

2) The Nrf2 protein expression by western blot is a valid methodology, however, as it is a transcription factor, an increase in expression does not necessarily reflect an increase in its activity, other techniques (such as luciferase) show better modulation.

Author Response

Point 1: Lines 75-77 of the introduction: the letter is enlarged.

Response 1: We apologize for our mistake. We have resized the enlarged letters to the same font size as the full manuscript and have made careful modifications on the original manuscript. All changes made to the text are in red color.

Point 2: The Nrf2 protein expression by western blot is a valid methodology, however, as it is a transcription factor, an increase in expression does not necessarily reflect an increase in its activity, other techniques (such as luciferase) show better modulation.

Response 2: This is indeed a very constructive question. As we explained in the introduction, Nrf2 is considered to be an important regulatory factor for ferroptosis. The activity of Nrf2 is rigorously regulated by Keap1. Keap1 not only passively isolates Nrf2 from the cytoplasm but also plays an active role in targeting Nrf2 for ubiquitination and proteasomal degradation. Under normoxic conditions, Nrf2 binds to Keap1 and continues to be inactivated by ubiquitination and degradation in the proteasome. Once the body is in oxidative stress, or if there are a large number of electrophiles or cytotoxic agents, Nrf2 is released from the Keap1 binding site and rapidly transferred to the nucleus, subsequently interacting with the antioxidant response element (ARE) in the promoter region of the target gene and then activates the transcriptional pathway to balance oxidative stress and maintain cellular redox homeostasis. Therefore, we detected the protein expression of Nrf2 and Keap1 by western blot and found that gastrodin could increase the protein expression level of Nrf2 and reduce the protein expression of Keap1, indicating that gastrodin may activate the Nrf2/Keap1 pathway by promoting the nuclear transfer of Nrf2. This was also confirmed by immunofluorescence assay. However, we did not detect the change of Nrf2 activity by dual luciferase assay or colorimetric method (Nrf2 Transcription Factor Assay). In the future studies, we will measure the activity of Nrf2 in hippocampus tissues or in hippocampal neurons according to your opinions and suggestions. Thank you again.